# 'Alone in a Crowd': Teacher-Level and Pupil-Level Hidden Curricula and the Theoretical Limits of Teacher–Pupil Relationships

Daniel Whittaker [ID]

Institute of Education, University of Worcester, Worcester WR2 6AJ, UK; d.whittaker@worc.ac.uk

**Abstract:** This essay aims to explore the theoretical limitations that hidden curricula—hidden normative values, beliefs, and knowledge that are often considered problematic—place on our understanding of teacher–pupil relationships. It applies Habermas' theory of communicative action—synonymous with mutual understanding and predicated on his concept of the lifeworld—to analyse hidden curricula. It finds that hidden curricula limit teachers' comprehension of teacher–pupil relationships dependent on pupils' responses to teacher-level hidden curricula. Where they respond with compliance, pupils limit expressions of their subjectivity; conversely, where they reject teacher-level hidden curricula, pupils' subjective lifeworlds are already disrupted by them. Both responses impede teachers' understanding of teacher–pupil relationships. In addition, pupil-level hidden curricula, which are often asymmetrical and oriented in response to teacher-level hidden curricula, present another barrier to teachers unveiling hidden curricula and the subjectivities of teacher–pupil relationships. In effect, pupil-level hidden curricula render teachers 'alone in a crowd'. Finally, I argue that systematically examining hidden curricula represents strategic action—communicative action's counterpoint—and colonisation of pupils' lifeworlds. While hidden curricula present significant theoretical limitations to understanding teacher–pupil relationships, teachers might use this as a 'pedagogical hinge', freeing them from the unknowable and uncontrollable to a more practical view of teacher–pupil relationships.

**Keywords:** teacher–pupil relationships; school; classroom; hidden curriculum; hidden curricula; Habermas; communicative action; lifeworld; intersubjectivity



## 1. Introduction

Teacher–pupil relationships have long been considered a core aspect of effective school teaching and learning [1–3], and positive teacher–pupil relationships are associated with positive educational outcomes for pupils [4–7], pupil engagement [8], and school attendance [9]. As a result, they also feature prominently within the UK's educational policy landscape [10–12]. While policymakers seek to take advantage of teacher–pupil relationships [10–13], a key question remains: what are the limits of teacher–pupil relationships? To evaluate their potential as educational levers and as research objects, we need to understand their contextual boundaries and theoretical limitations [14,15]. This is the aim of this paper. By examining the concept of hidden curricula using Habermas' communicative action and lifeworld concepts, this paper argues that pupil-generated hidden curricula—and pupil responses to teacher-level hidden curricula—render teachers 'alone in a crowd' [16] and present a theoretical and practical limit to teacher–pupil relationships.

### 1.1. Hidden Curricula

Jackson [16] was the first to study the hidden curriculum systematically; his functionalist, atheoretical approach described the hidden curriculum as a productive means of organising classroom activities and behaviours [17]. As Dickerson [18] (pp. 48–49) notes:

> [Jackson] stressed that while it was unstated, the hidden curriculum was clearly taught and learned in school, and was, in fact, a systematic and powerful way

to teach students about themselves, their place in the world, and what learning might be like.

To Jackson [16], the hidden curriculum presents 'three facts of life [. . .]: crowds, praise and power' (p. 38) through which pupils learn to negotiate the complex power dynamics between the competing demands of their teachers and their peers. To succeed, they must learn how to elicit positive evaluations, how to suspend their innermost desires and 'how to be alone in a crowd' (p. 42).

In contrast, Giroux and Penna define the hidden curriculum as the 'unstated norms, values and beliefs that are transmitted to students through the underlying structure of meaning in both the formal content as well as the social relations of school and classroom life' [19] (p. 22). Through a critical lens, they argue for unveiling and challenging the hidden curriculum in a way that 'penetrates the functional relationships that exist between the institutions of the schools, the workplace and the political world' (p. 38).

### 1.2. The Distinction between Teacher-Level and Pupil-Level Hidden Curricula

While 'hiddenness' is explicit within the term 'hidden curriculum', it does not indicate who the curriculum is hidden from or if the hiding is intentional: Portelli's analysis demonstrates that curricula hidden intentionally (or unintentionally) are feasible logical possibilities [20]. Another possibility is for hidden curricula that are not teacher-generated. This presents a problem: it implies that aspects of the curriculum are hidden from pupils, teachers and researchers alike.

Skelton's definition of the hidden curriculum accounts for such pluralities [21] (p. 185):

> The hidden curriculum is that set of implicit messages relating to knowledge, values, norms of behaviour and attitudes that learners experience in and through educational processes. These messages may be contradictory, non-linear and punctuational and each learner mediates the message in her/his own way.

Because this definition accounts for the fluid interactions between individuals and hidden curricula, it is a suitable definition to adopt for the present study, which is concerned with teacher–pupil relationships.

Indeed, Skelton's definition allows for the pupil-level hidden curricula observed by other researchers [21–25]. Pupil-level hidden curricula are similar to teacher-level hidden curricula in that they are also defined by implicit messages, values and norms but different in that they exist between pupils and are hidden from teachers. Whereas teacher-level hidden curricula are often institutionally generated and oriented, pupil-level hidden curricula are perpetuated by pupils, mediating pupils' educational experiences and learning. For example, pupil-level hidden curricula might concern norms of the aesthetics of dress and hair, or the level of academic engagement pupils expect from each other [21]. They can also arise in response to the formal curriculum and teacher-level hidden curriculum through peer-mediated attitudes to 'cheating, completing homework, striving for academic success, exhibiting attitudes towards teachers' [22] (p. 127). As such, pupil-level hidden curricula might support or even conflict with teacher-level hidden curricula. This conflict might manifest in unsanctioned peer talk and 'off-task' behaviour, where pupil-level hidden curricula promote behaviours that subvert classroom processes [23]. Alternatively, the conflict might encourage the promulgation of racist and sexist beliefs that run counter to those in the teacher-level hidden curricula [24]. However they manifest, pupil-level hidden curricula present a challenge to teachers because they are both hidden from them and influence the quality of the classroom climate [25].

### 1.3. Hidden Curricula as Problematic

Despite some authors advocating hidden curricula as legitimate forms of learning [16,26], they are more often presented as problematic, especially for critical theorists [17,18,21] who urge us to unveil their covert, normative effects [19,20,27]. For Huttunen, the hidden curriculum is a form of indoctrination, which entails 'infiltrating (drilling, inculcating, etc.)

concepts, attitudes, beliefs, and theories into a student's mind by-passing her free and critical deliberation' [28] (p. 1), and Portelli contends that 'teachers, as responsible persons, have the moral responsibility to diminish undesirable, unintended consequences to the extent that this is possible' [20], (p. 351). Thus, the hidden curriculum is a crucial concern for anyone looking to understand the nature of the teacher–pupil relationship, especially relationships oriented toward equity and symmetry. While space does not permit us to evaluate the imperative to unveil the hidden curriculum here, the extent to which it is possible *is* within this paper's purview; one tool apt for such an examination is Habermas' theory of communicative action, which I will explore in more detail in the next section.

### 1.4. Communicative Action and Teacher–Pupil Relationships

In short, Habermas' theory of communicative action concerns the pursuit of mutual understanding and the theoretical conditions that make such a goal possible [29] and has been applied to analyse various aspects of teacher–pupil relationships. On a theoretical level, communicative action has been proposed as a tool to evaluate the democratic aims and means of educational practice [30] and to overcome the communicative deceit that limits emancipatory practice [31]. Communicative action has also been applied to practical teacher–pupil relationship issues such as classroom feedback processes [32,33], pupil and teacher talk [34], assessment practices [35] and the increased juridification of pupil–pupil and teacher–pupil relationships [36,37].

I will draw on the similar affordances of communicative action to examine hidden curricula in this study. Using Habermas' concept of the lifeworld [29], I aim to explore teacher-level and pupil-level hidden curricula to illuminate the limits of our understanding of teacher–pupil relationships. The next section will explain the theoretical framework I will use to undertake such an analysis. In later sections, I argue that hidden curricula disrupt and distort pupils' and teachers' lifeworlds and their ability to act communicatively through teacher–pupil relationships. Finally, I make recommendations for further research and argue that the significant limitations hidden curricula place on our ability to understand teacher–pupil relationships can be considered a liberating 'pedagogical hinge' [38].

## 2. Methodological Framework

Habermas' theory of communicative action can be summarised as action towards mutual understanding [29]. It is predicated on Habermas' concept of the lifeworld—'the background resources, contexts, and dimensions of social action that enable actors to cooperate on the basis of mutual understanding: shared cultural systems of meaning, institutional orders that stabilise patterns of action, and personality structures acquired in family, church, neighbourhood, and school' [39]—and stands in opposition to strategic action which is action directed towards an outcome through means of manipulation [27,28]. As such, communicative action's position opposite to the covert, one-sided meaning of the hidden curriculum [21] makes it a suitable heuristic to examine its nature.

### 2.1. Communicative Action as Pedagogy

On that basis, communicative action is an apt model to examine the possibility of the imperative to unveil the hidden curriculum because of its clear focus on social action towards mutual understanding [40]. Many authors have proposed this pedagogical approach, albeit in modified forms. Huttunen advocates 'communicative teaching' as a 'simulation of communicative action, a simulation of a free and equal discourse' where teachers negotiate meaning with students [28] (p. 12). It differs from communicative action in that Huttunen sees the teacher/student relationship as asymmetrical: students are not as communicatively competent as their teachers, a key condition for communicative action. In contrast, Biesta's approach proposes 'practical intersubjectivity', which treats the pedagogical relationship as symmetrical because 'pedagogical action is not considered to be a one-way process in which meaning is transferred' so can be 'thought of as a co-constructive process in which meaning is produced' [30] (p. 312).

This contrast in approaches highlights a key problem for applying communicative action in education: communicative competence of all participants is an unavoidable precondition. Students—especially young students—lack this communicative competence, which, we may argue, is why they require an education in the first place. If we consider education as an initiation into that which society deems worthwhile [41], the teacher is the prime initiator of the skills, knowledge and dispositions that might constitute such an education. The difference in knowledge, power and communicative competence between students and teachers renders communicative action an unsatisfactory and unrealistic pedagogy for direct application/praxis.

*2.2. Communicative Action as a Heuristic*

Despite its limitations as a direct pedagogical approach, the theory of communicative action is an effective thinking tool. For Morrison, 'Habermas's work was problematical in its several elements but that it might have instrumental, heuristic value in establishing a set of principles with which to interrogate the curriculum' [27] (p. 183). He justifies 'the inclusion of Habermasian principles in a commentary on education' (p. 365) on the grounds that they are 'consciousness-raising' (p. 365), offer 'interesting and empowering ways of analysing and approaching education' (p. 365) and start 'where people are rather than were they ought to be' (p. 366).

Communicative action's principal utility derives from the primacy it gives to reaching mutual understanding for all members of communication communities. Early forms of Habermas' communicative action assumed an 'ideal speech situation' requiring that every person wishing to contribute has an opportunity to propose and challenge what they honestly believe to be true as well as holding themselves and others to account for their behaviour, whereas his later work subsumes these ideas into 'universal principles of argumentation' [28]. Examining the hidden curriculum against these necessary preconditions for communicative action enables a view of participation: it will foreground who is subject to the hidden curriculum, who the curriculum is hidden from, who contributes to its creation and to what extent all of these are intentional. Indeed, I propose that examining hidden curricula through Habermas' communicative action will illustrate the extent to which they are accessible to teachers, students and researchers.

*2.3. The Lifeworld*

In his theory of communicative action, Habermas uncoupled the concepts of system and lifeworld [29]. According to Baxter [42] (pp. 45–46):

> Communicative action, [Habermas] acknowledges, takes place within a social context—a context that he [...] calls the "life-world". The phenomenological tradition has conceived of the life-world as the "horizon" within which individuals seek to realise their projected ends.

Thus, the lifeworld and its 'horizons' offer further utility in examining the hidden curriculum and teacher–pupil relationships.

Habermas further divided the lifeworld into subjective, intersubjective and objective worlds: the subjective world refers to individuals' experience; the intersubjective world refers to relationships between people; the objective world refers to an objective, concrete reality [29]. Because these worlds are deeply intertwined, they can never be fully examined in isolation; however, Habermas' notional conceptual separation affords us the ability to understand in more detail how the system and lifeworld influence each other and to locate barriers to communicative action [35,43].

Communicative action and strategic action are aligned with the lifeworld and system, respectively [29], a distinction that enables an understanding of the origin and effects of hidden curricula. Hidden curricula, it may be argued, belong in the category of strategic action: if strategic action is instrumental action towards an outcome, hidden curricula—'that set of implicit messages relating to knowledge, values, norms of behaviour and attitudes that learners experience in and through educational processes' [21] (p. 185)—is a clear

orientation towards outcomes without recourse to understanding (as evinced through its hiddenness).

Habermas argued that systems can colonise lifeworlds and that the rise of economic and politico-legal systems narrows the scope of pedagogical action available to teachers [29]. Rossiter's study offers an example of this: he found that the rise of standardised testing in the US—an educational imperative with systemic origins—profoundly influenced relationships in the lifeworld of the classroom [35]. Indeed, according to Rossiter, educational imperatives are systemically originated and 'can only be accomplished through lifeworld functions, that is, through the communicative procedures in which teachers and students engage in critical inquiry in social contexts' (p. 38). For Rossiter, then, education is predominantly situated in the intersubjective world and thus concerned with relationships between people. Therefore, if we assume that hidden curricula are systematically initiated and education predominantly exists in the intersubjective world, it becomes possible to examine how (or indeed whether) the system and the lifeworld interact through hidden curricula, offering an insight into the theoretical nature of teacher–pupil relationships.

### 2.4. Operationalising Communicative Action

For the purposes of this study, the research subject is not a particular hidden curriculum, situated in a particular context. Following Morrison's argument that Habermasian principles are best applied as a 'commentary on education' and its principles [27] (p. 365), I will examine the hidden curriculum concept as a theoretical entity empirically observed by other researchers.

Because communicative action is predicated on the concept of the lifeworld, this is the most appropriate place to start examining the nature of hidden curricula. From there, I consider both the teacher's and pupil's access to the hidden curriculum and its impact on teacher–pupil relationships.

## 3. Hidden Curricula and Lifeworlds

Insofar as they can be determined, lifeworlds are situationally dependent: 'The lifeworld forms the setting in which situational horizons shift, expand, or contract. It forms a context that, itself boundless, draws boundaries' [29] (p. 152). Teachers and students exist in different lifeworlds at different times, and although the nature of each lifeworld is unique, they overlap considerably. For example, a pupil's lifeworld at home will be shaped by their subjectivity, which they will bring into the classroom with similar effects. In a classroom of one teacher and 30 pupils, 31 subjectivities will be reflexively influencing each other. As the subjective, intersubjective and objective worlds are inexorably intertwined, these individual subjectivities will profoundly influence the intersubjective nature of the classroom.

Where hidden curricula are concerned, these overlapping lifeworlds lead to 'different versions of the hidden curriculum being realised in different settings' [40] (p. 184). At the heart of this is:

> The assumption that people, including students, are active participants in the creation and interpretation of their social environments and action. But students are not independent agents; they are shaped by history and culture, and by the immediate social relations and practices of schooling. [44] (p. 52)

A teacher cannot experience what a pupil experiences; they do not have access to that aspect of the lifeworld. However, pupils do give clues to their subjectivity. Observing pupils' body language and semi-private utterances, Nuthall inferred how some pupils experience the classroom. However, observing these small manifestations of subjectivity is incredibly resource-intensive [45]. This presents the first fundamental limitation of teacher–pupil relationships: even if we accept that it is possible to understand pupil subjectivities adequately, for a teacher or researcher to gather this data for an individual pupil in the moment would be unfeasible; to do this for a whole class on an ongoing basis would be a practical impossibility.

*3.1. Subjectivity and Pupil Responses*

Indeed, to observe manifestations of individual subjectivity also relies on the teacher's (and researchers') own subjective interpretations. This situational intersubjectivity is illustrated by Alerby, who contends that 'a physical room, such as a classroom, is perceived in very different ways depending on whether one is a child or an adult, a student or a teacher, and depending on what role one has in this context' [46] (p. 16). This foregrounds an important arena of intersubjective conflict between the teacher and the student: 'The teacher, as well as the physical room, can in most cases only control the physical presence of the students, not where they mentally are' [46] (p. 17). Therefore, hidden curricula can only be applied intentionally insofar as pupils' subjectivities allow. To some extent, this undermines the value of a teacher recognising the effect of the intentionally hidden curriculum. Alton-Lee, Nuthall and Patrick's study illustrates this well: their description of a teacher whose intention was to 'increase the children's tolerance of cultural differences' [23] (p. 80), but whose 'hidden curriculum of differential cultural valuing was more powerful in the lesson we selected than his official agenda' (p. 80) demonstrates how an unintentional hidden curriculum can supersede and contradict the intentional curriculum.

The differences in subjective perception between pupil and teacher may, in part, be explained through a psychological perspective. Jackson proposes that 'all students probably learn to employ psychological buffers that protect them from some of the wear and tear of classroom life' [16] (p. 49). As such, they learn how 'to disengage, at least temporarily, their feelings from the actions' (p. 43) (and vice versa) and learn 'how to be alone in a crowd' (p. 42). The corollary of students successfully disengaging is reduced manifestations of pupil subjectivity and, therefore, reduced opportunities for teachers to observe the effects of the hidden curriculum.

Conversely, where students are unsuccessful in psychologically adjusting to classroom norms, their subjective response may be clearly demonstrated. Woods categorises six adaptations to classroom control aspects of hidden curricula: conformity, ritualism, retreatism, colonisation, intransigence and rebellion [47] (pp. 71–72). Through manifestations of rebellion and intransigence, pupils' behaviour illustrates an overt rejection of hidden curricula. Nevertheless, whether pupils respond with compliance—minimising manifestations of their subjectivity—or whether they reject the hidden curriculum, both represent pupil-oriented limitations to the teacher–pupil relationship: the former to teachers' intersubjective understanding and the latter by disrupting pupils' own subjective lifeworld.

Habermas' validity claims offer another instructive angle to explore pupils' responses to hidden curricula. According to Habermas [48], 'anyone acting communicatively must [. . .] raise universal validity claims' (p. 22), which can be summarised as:

- Comprehensibility, or 'uttering some intelligibly';
- Truth, or 'giving (the hearer) something to understand';
- Truthfulness, or the speaker's desire to express their intentions 'truthfully' and understandably;
- Rightness, or seeking mutual agreement 'with respect to prevailing norms and values'.

When pupils comply with hidden curricula, their reduced manifestations of pupil subjectivity do not satisfy the truthfulness claim. However, pupils' rejection of hidden curricula signifies their unwillingness to reach a mutual agreement and, therefore, does not satisfy the rightness claim. In both cases, the validity claims necessary for communicative action are not met. Indeed, the very hiddenness of hidden curricula denies truthfulness and possibly comprehensibility, depending on whether participants recognise or respond to them.

When mutual interpretation fails, participants break off communication altogether or switch to strategic action [48]. Thus, hidden curricula constitute strategic action. Because strategic action is directed towards an outcome through manipulation [27,28], hidden curricula render any teacher–pupil relationship the site of this manipulation. While teacher-level hidden curricula presuppose teachers' denial of communicative action, pupil-level hidden

curricula—to which we now turn our attention—presuppose a denial of communicative action from the pupils.

### 3.2. Pupil-Level Hidden Curricula

Another way hidden curricula limit the understanding of teacher–pupil relationships is teachers' inability to access pupil-level hidden curricula. Perhaps the first study to demonstrate this was Snyder's exploration of the hidden curricula at the Massachusetts Institute of Technology [49]. He found a distinct but 'covert student culture which represents such an important section of the hidden curriculum' (p. 25); it normatively influenced behaviour between students and their peers as well as between students and lecturers. As such, it created 'a political shift, or a transfer of power from administrators to students, with the latter sharing the major responsibility for determining rules and social conduct' (p. 23) that often required students to choose patterns of behaviour from one hidden curriculum that would put them in conflict with the other.

Nuthall [45] and Alton-Lee, Nuthall and Patrick, J. [23] found similarly conflicting hidden curricula in primary classrooms in New Zealand. Both authors found that pupil interactions were mainly invisible to teachers—a product of pupils' effectiveness in concealing their behaviour to avoid conflict with the teacher-level hidden curriculum. These semi-private pupil interactions are fraught with normative expectations, constituting a pupil-level hidden curriculum. Because 'transgressing peer customs may have worse consequences than transgressing the teacher's rules and customs' [45] (p. 84), hiding their interactions is necessary to negotiate successfully both hidden curricula. Perhaps most concerning for teacher praxis is the observation that pupils learn most of the content presented to them through peer interactions:

> The important insight that comes from these exchanges is that much of the knowledge students acquire comes from their peers, and when it does, it comes wrapped inside their social relationships. [45] (p. 93)

The hidden curricula wrapped in these social relationships are subject to the same power imbalances as teacher-level hidden curricula. For example, Rietveld found that although teachers thought pupils with Down's Syndrome were well integrated in their classrooms, they were ignorant of pupil-level and teacher-level interactions promoting discourses damaging to the self-concept of the pupils with Down's Syndrome; pupils would 'baby' them and quietly ridicule them and their behaviours [50]. Thus, these examples illustrate not only how a teacher's ignorance of pupil-level hidden curricula is a major hindrance to their intersubjective understanding of teacher–pupil relationships but also that this ignorance is a necessary feature of both teacher–pupil and pupil–pupil relationships.

### 3.3. Alone in a Crowd: Teachers' Lifeworld Access

The previous chapter exemplified an unfortunate reality for teachers and researchers attempting to uncover and deal with hidden curricula and their effects: teachers have very little access to much of their classroom's lifeworld. Pupil-level hidden curricula are an intersubjective reality that profoundly influences pupil learning and self-concept. Despite a willingness to enlighten and emancipate themselves and their pupils, the critical teacher is rendered impotent by an inability to access this intersubjective world that is simultaneously a reaction to the teacher's hidden curriculum and intentionally hidden from them. The cruel irony is that teachers become 'alone in a crowd', the reverse of Jackson's observation of the expectation of teacher-level hidden curricula [16] (p. 42).

Being so alienated from so much of the classroom's lifeworld makes communicative action a practical impossibility. By being excluded from the pupils' intersubjective world, a teacher's subjective experience is isolated from their pupils' subjective experience; consequently, the teacher's conception of the effects of their practice becomes potentially problematic. Rietveld's study offers a case in point: the teachers saw a very different reality from what actually existed [50]. Communicative action assumes communicative competence and a forthcoming honesty to express oneself truthfully; because pupil-level

hidden curricula are either intentionally hidden to reduce conflict with the teacher-level hidden curriculum or unintentionally hidden through a lack of communicative competence, they present significant barriers to mutual understanding.

## 4. Hidden Curricula and Intersubjective Symmetry

Applying Habermas' theory of communicative action [29] to the hidden curriculum has revealed a few critical limitations to the potential for teachers and researchers to comprehend and influence teacher–pupil relationships. First—depending on how pupils respond to the teacher-level hidden curriculum—teachers are severely limited on how they can understand their pupils: if a pupil complies with this hidden curriculum, they will limit manifestations of their subjective lifeworld, but if a pupil rejects it, their subjective experience of the teacher will be mediated. Beyond this, even if it were possible for a teacher to understand the teacher–pupil relationship by unveiling the hidden curriculum, the intensive nature of seeking to understand the subjective lifeworld of each child would render it practically impossible. To compound this, the existence of pupil-level hidden curricula—intentional or otherwise—veil teachers' understanding of their relationship with pupils.

### 4.1. Intersubjective Symmetry

Pupil-level hidden curricula also indicate a possible indirect symmetry to the teacher–pupil relationship. If we assume that pupil learning is mediated through pupils' relationship with their teachers and peers, as indicated by Nuthall [45], and that pupil-level hidden curricula are an intersubjectively negotiated response to teacher-level hidden curricula, as indicated above, it becomes possible to see that one is reflected in the another. I argued earlier that hidden curricula may be considered strategic action because of their orientation toward systems and organisation. However, just as teacher-level hidden curricula may be oriented in response to institutional demands, they could equally be considered a response to pupil-level hidden curricula. Thus, both levels of hidden curricula are practically, reflexively related to each other.

This is consistent with Biesta's [30] (p. 30) notion of practical-intersubjectivity where the 'practical' element:

> means that human intersubjectivity should first of all be understood in terms of action. "Practical intersubjectivity" thus designates a structure of communicative relations "that arises and takes form in the joint activity of human subjects to achieve ends set by their life needs". [40] (p. 30)

Because the pupil-level and teacher-level hidden curricula are mutually responsive to each other's ends and needs, the intersubjectivity takes on an almost symmetrical, circular pattern.

To what extent any symmetry is theoretically possible and empirically verifiable is unclear. One complication is that while we can reasonably assume that teacher–pupil relationships are built on an imbalanced power relationship, we can also reasonably assume that pupil–pupil relationships are built upon similarly unequal and unstable power relationships. Nuthall's study illustrates this imbalance, illustrating pupils' knowledge of and response to their place in the classroom's power structure:

> Jim's sensitivity to Paul's and Tilly's comments results in him reacting angrily to both of them. Getting help from his peers is a minefield for Jim, who has limited reading and spelling ability. During this interchange, the students play out and develop their respective roles in relation to the knowledge needed to succeed in the class. The exchange confirms that Paul knows most of the required knowledge, that Jim is treated as a fool, that Koa is a helpful ally, and so on. [45] (p. 89)

Thus, while the pupil-level hidden curriculum is responsive to and potentially reflective of the teacher-level hidden curricula (and vice versa), the former is—similarly—a product of a shifting network of peer power. As such, the instability of pupil-level and teacher-

level hidden curricula is both a necessary condition of the teacher–pupil relationship as well as a severe limitation to understanding it.

*4.2. System/Lifeworld Colonisation*

So far, I have made the case that hidden curricula represent strategic action—aligned with the concept of the system—and that through pupil and teacher-level hidden curricula, the system and lifeworld interact with each other. For Habermas, one of the key interactions is the colonisation of the lifeworld by the system [29]. To what extent, if any, can it be said that these hidden curricula colonise the classroom lifeworld and how would this impact our understanding of teacher–pupil relationships?

Kemmis argues that colonisation occurs when:

> the imperatives of the economic and political-legal systems dislodge the internal communicative action, which underpins the formation and reproduction of lifeworlds, providing in its place an external framework of language, understandings, values and norm based on systems and their futures. Under such circumstances, the symbolic reproduction processes of the lifeworld [...] become saturated with a discourse of roles, futures and functionality, reshaping individual and collective self-understandings, relationships, and practices. [43] (p. 280)

Just as Kemmis argues, the strategic action of teacher-level and pupil-level hidden curricula saturate—or colonise—the classroom lifeworld with norms, values and understanding, making the mutual understanding of communicative action a practical impossibility.

Alternatively, it is possible to conceive of hidden curricula originating and operating within the lifeworld rather than colonising it. Our definition of hidden curricula as 'implicit messages relating to knowledge, values, norms of behaviour and attitudes that learners experience in and through educational processes' [21] (p. 185) corresponds closely to both the subjective and intersubjective worlds described by Habermas [29]. Through teachers' intersubjective position in relation to pupil-level hidden curricula and through their subjective experience of it, their lifeworld is inextricably linked to their pupils'. If we also consider that pupil-level hidden curricula originate in pupils' lifeworlds and recall Rossiter's argument that educational imperatives—the hidden curricula being one example—can only be accomplished in and through the lifeworld [35], we can propose that through hidden curricula, the lifeworld acts on itself.

Turning the action of a lifeworld back on itself exposes a flaw in the uncoupling of system and lifeworld underpinning the theory of communicative action. However, we can see the limitations of comprehending the teacher–pupil relationship, whether we consider the lifeworld as being colonised by, or originators of, hidden curricula. If we assume that hidden curricula originate in pupils' lifeworlds, we can assume that they are not accessible to teachers, precisely because—as we have already established—teachers cannot access a pupil's lifeworld. Conversely, if we assume that hidden curricula—an orientation toward systems and strategic action—colonise lifeworlds, we can also assume that the mutual understanding of communicative action—the antithesis of strategic action—is not possible. In addition, any systematic examination of the hidden curriculum and teacher–pupil relationship can also be considered—by definition—strategic action, rendering it a colonising process in and of itself.

## 5. Conclusions

In this paper, I aimed to explore the limitations of the teacher–pupil relationship by using Habermas' theory of communicative action as a heuristic tool to examine the hidden curriculum. The concept of the lifeworld offered great utility. Because the lifeworld represents the social context in which communicative action might occur, it also represents the horizon upon which it might be possible. However, hidden curricula obscure or distort this horizon for teachers and pupils alike, affecting our ability to understand teacher–pupil relationships.

Teacher-level hidden curricula present the first distortion, depending on how pupils respond to them. When pupils acquiesce to the teacher-level hidden curriculum, their acquiescence obscures teachers' ability to recognise and appreciate their pupils' subjective experience. However, when pupils reject teacher-level hidden curricula, their rejection itself represents a distortion of the pupils' lifeworlds. Thus, through the concept of the lifeworld, we may conclude that teacher-level hidden curricula present a significant barrier to our understanding of the teacher–pupil relationship. Whatever pupils' responses to hidden curricula, their presence distorts pupils' and teachers' validity claims and, therefore, precludes communicative action. Thus, hidden curricula constitute the teacher–pupil relationship as a site of strategic action and manipulation.

Pupil-level hidden curricula constitute another theoretical limit to our understanding of teacher–pupil relationships. By simultaneously mediating pupils' subjective experience while excluding teachers, they are inaccessible to teachers and researchers alike. In addition, any attempt to systematically examine pupil-level hidden curricula and pupils' experience of them requires colonising the lifeworld with the system, another manifestation of strategic action.

By applying Habermas' theory of communicative action to hidden curricula, this study has established that lifeworld access represents a key theoretical limitation to our understanding of teacher–pupil relationships. However, rather than closing off further inquiry into the hidden curriculum and teacher–pupil relationships, this finding offers a point of departure to apply Habermas' other concepts and tools.

One promising avenue for analysis might consider the hidden curricula and teacher–pupil relationships in relation to communicative competence [29]. Siljander argues that while communicative action ostensibly requires equality, anyone participating in any communicative or educational event will likely have different and unequal communicative competencies [31]. How might the communicative capacity of the youngest children in our schools differ from those in higher education, for example? How might these differences impact the nature of hidden curricula and teacher–pupil relationships in these educational phases?

While this study briefly explored the hidden curriculum in terms of Habermas' four validity claims necessary for communicative action [48], another direction for potential research might consider the implications of these for trust, authority and mutual respect within the educational process. Also, while hidden curricula represent strategic action and thus colonisation of the lifeworld by the system, we have not explored how this colonisation might differ in different phases of education and different subjects. For example, in Toward a Rational Society, Habermas states his desire to subordinate technical rationality with democratic processes and lifeworld experiences [51,52]. In contemporary society, how might the hidden curricula of university mathematics and nursery education, for example, compare? How will these particular hidden curricula impact the teachers, pupils, students and their relationships?

A final recommendation would be to explore how a teacher's personality might influence teacher–pupil relationships. For example, how might an orientation toward authoritarianism shape the hidden curriculum? An analytical framework such as Adorno's The Authoritarian Personality might offer a productive lens to illuminate this hidden corner in our understanding of the hidden curriculum [53].

While this essay explored the theoretical limits of the teacher–pupil relationship, it offers a way for teachers to evaluate their agency in building and maintaining classroom relationships. The suggestion that the hidden curriculum teaches pupils how to be 'alone in a crowd' [16] (p. 42) exposes a cruel irony: teachers, too, are subjected to a similar alienation from their lifeworld. This is potentially problematic: cynically acquiescing to impotence could result in teachers retreating in pursuits of communicative action and mutual understanding. To counter this, I propose that it presents an opportunity to reflect on an emerging sense of symmetry in the classroom. As a result of teachers' isolation from pupil-level hidden curricula, there is an opportunity to consider an orientation to

praxis that places them in an alternative, relatively inaccessible network of power and relationships. This echoes Sojot's concept of a 'pedagogical hinge' where:

> The hinging moment [. . .] elucidates the sense of pedagogical becoming: it occurs in the relationship of experiencing the learning self and of the growing awareness of the learning self through its interaction with the environment or object. There is an element of letting go and the recognition of the inability to control the outcome or affect of an environment or object presupposed as an educational tool. [38] (p. 899)

> Whether teachers are encouraged to engage with teacher–pupil relationships through an instrumental lens as directed by policy/system [10–13] or as part of a critical imperative [18–21], teacher-level and pupil-level hidden curricula remain significant limiting factors. Thus, for the teacher–pupil relationship, hidden curricula represent pedagogical hinges defined by the unknowable and the uncontrollable. As Sojot suggests [38], this offers an opportunity for teachers to 'let go' and reclaim the realisation that not every teacher–pupil relationship is a product of their intention, skill or effort.

**Funding:** This research received no external funding.

**Institutional Review Board Statement:** Not applicable.

**Informed Consent Statement:** Not applicable.

**Data Availability Statement:** Not applicable.

**Conflicts of Interest:** The author declares no conflicts of interest.

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
