# Peer review of "‘Alone in a Crowd’: Teacher-Level and Pupil-Level Hidden Curricula and the Theoretical Limits of Teacher–Pupil Relationships"

_education, doi:10.3390/educsci14050477_

Round 1

Reviewer 1 Report

Comments and Suggestions for Authors

A well-written essay that could have been made stronger by referencing Habermas's work that deals directly with issues of education, namely "Toward a Rational Society: Student Protest, Science and Politics." The article could have also benefited by engaging with theories of "The Authoritarian Personality," which also comes from the Frankfurt School and examines the relationship between teachers and students. 

That said, the article does a great job of placing the theory of communicative action within the context of teaching and learning and viewing these as essential to meaningful mutual engagement between teachers and students.

Reviewer 2 Report

Comments and Suggestions for Authors

Dear authors,

Congratulations on your interesting research.

The research is engaging and addresses a significant topic. It presents an essay on the hidden curriculum and the teacher-pupil relationship analyzed from the perspective of Habermas' theory of communicative action.

I recommend that the article be accepted pending minor revisions . My reason for this recommendation is that the presented research is methodically correct in all aspects and presents the results in a comprehensive, structured scientific manner.

However, the following suggestions are offered for the consideration of the authors:

From the outset, including in the abstract and introduction, the aim of the analysis or essay should be directly stated. Conclusions drawn should align with and relate back to this stated purpose/aim.

The abstract reports the outcomes of the research. A short motivation of research outlines the importance of the problematic. Some sentences from the abstract are difficult to follow and should be reformulated and shortened to improve the clarity of the text. An example would be "It finds that teachers' comprehension of the teacher-pupil relationship is limited by hidden curricula which may be dependent on how pupils respond to teacher-level hidden curriculum: if pupils reject the hidden curricula, the teacher-pupil relationship is already disrupted by it, but if pupils comply, teachers are less likely to understand pupils' subjective experience of it."

In the introductory section, where different definitions of the Hidden Curriculum are outlined, it's essential to elucidate the two perspectives analyzed - the Hidden Curriculum at the teacher level and the student level - providing examples for clearer comprehension. Additionally, a more detailed overview of Habermas' Theory of Communicative Action should be presented, detailing its application in the analysis of teacher-student relationships.

The theoretical framework is well written and present the analysed concepts in a comprehensive manner. All the presumptions and affirmations are supported by many up-to-date references.

In an academic essay, especially one that leans towards empirical research using a title like "Materials and Methods" is appropriate for sections where you describe the methodologies, tools, frameworks, and materials used in your analysis or research. The conventional use of "Materials and Methods" is more common in scientific research papers than in theoretical essays. In such a case, a more fitting title for your section might be "Methodological Framework" or "Theoretical and Analytical Framework”.  

Another promising avenue for analysis might involve exploring validity claims: Investigating how the four validity claims (truth, rightness, sincerity, and comprehensibility) manifest in the teacher-pupil relationship. The implications of these claims for trust, authority, and mutual respect within the educational process should also be discussed.

The conclusions should be restructured to clearly demonstrate the objective and the way it was accomplished. Additionally, in the conclusions section, it would be beneficial to include implications for future research and practice,  proposing directions for further investigation into the teacher-pupil relationship through Habermas' framework and its implications for educational practices.

Congratulations to the authors!

Sincerely,
